# The Impacts of Online Clothes Short Video Display on Consumers' Perceived Quality

**Rong Ma [1], Bingjia Shao [2,3], Jiaqi Chen [4] and Dan Dai [5,\*]**

[1] School of Economics and Management, Yunnan Agricultural University, Kunming 650201, China; 2011019@ynau.edu.cn

[2] School of Economics and Business Administration, Chongqing University, Chongqing 400044, China; shaobingjia@cqu.edu.cn

[3] Chongqing Key Laboratory of Logistics at Chongqing University, Chongqing 400044, China

[4] School of Economics and Business Administration, Central China Normal University, Wuhan 430079, China; Jiaqi@cqu.edu.cn

[5] Institute of Quality Development Strategy, Wuhan University, Wuhan 430072, China

\* Correspondence: dandai@whu.edu.cn; Tel.: +86-157-2326-1867

**Abstract:** In the context of the rapid development of online shopping, enterprises and consumers realized the importance of an effective online short video display. However, academics rarely explored the content of a short video display and its impacts on consumers' perceptions. This paper discusses the impacts of this display form on consumers' perceived quality via questionnaires and scenario experiments based on the existing literature and theories. A short video of clothes was the main research object. We used an experimental design that included 2 (time: long, short) × 2 (display perspectives: overall, specific) × 2 (scenarios: outdoors, indoors) groups of experiments, for a total of eight groups of scenario experiments, to verify our hypotheses. The results reveal that the short video display makes consumers have a higher perceived quality compared with the long video display. Compared with a video that only includes an overall view of the product, a video that provides both an overall view and the details of the product makes consumers have a higher perceived quality. Consumers were found to have a higher perceived quality of products shot outdoors compared with products shot indoors.

**Keywords:** online clothes retail; short video; perceived quality

## 1. Introduction

E-business is rapidly developing, as combining online businesses and offline businesses promotes interaction between consumers and businesses and inspires innovation in retail businesses around the world. However, the virtual nature of an E-business, to some extent, makes consumers lack trust, which negatively affects the development of E-businesses [1]. Consumers face greater risks when shopping online, because they cannot physically touch products, and they are unable to perceive a product's quality; that is, there is uncertainty in their perception of a product's quality [2,3]. Online retailers are attempting to elevate consumers' perceptions of products' quality through optimizing displays of product information by using a variety of tools to comprehensively represent products' appearance, materials, and related attributes [4,5]. The short video is a rapidly developing form of content marketing. It not only provides a smooth and vivid visual experience of products [6], but also expands the marketing space. Online retailers may utilize short videos to increase the order conversion rate.

To date, studies related to the display of product information primarily focused on the impacts of interactivity and vividness on consumers [7–11]. Yoo and Kim believe that the images and text in a product display, which stimulate perception, are the primary information sources with which consumers make purchase decisions [12]. A static display makes consumers have a positive perception of searched-for items; a dynamic display makes consumers have a positive perception of experienced products [13]. Imagination plays a key role in the making of purchase decisions when consumers are unable to physically touch products [14]. In the context of online shopping, when information on a product is missing or ambiguous, consumers may be suspicious [15], which may affect their perceptions and imagination [16], lessening their intention to purchase. According to multi-sensory interaction and integration theory, E-business product displays, via hearing sensations and visual sensations, stimulate consumers' tactile sensory associations to promote virtual tactile sensations [17,18]. Wirth et al. [19] point out that the more sensory information that online retailers provide and the more that this information matches with consumers' perceptions, the easier it is for consumers to have a better experience of virtual shopping. Taking imagination as an important subjective experience of online shoppers into consideration, the vividness and amount of information included in a product display are important in elevating consumers' perceptions of the product's quality.

Compared with a static display, a short video display that includes visual and auditory sensory information and features multi-sensory interaction has advantages in stimulating consumers' imagination to enhance consumers' perceptions of products' quality [20–22]. Oru et al. [23] took incentive for information processing as a mediating variable from the perspective of consumers' imagination, and they explored the impacts of video displays' presentation and features on consumers' information processing, attitude toward products, and purchase intentions. Flavián et al. [24] found that a video display can influence consumers' perceptions of E-business products, purchase channel preferences, and purchase intentions in two respects: ease of imagining products and satisfying demand for tactile sensations. Guo et al. [25] developed a theoretical model for the impacts of short video displays on consumers' purchase intentions based on the "S–O–R" (stimulus–organism–response) model. The authors found that the usefulness and comprehensiveness of information positively influence consumers' virtual tactile sense, enjoyment, and trust.

To date, academics focused on exploring the characteristics of short videos and their impacts on consumers' purchase intentions as compared to static displays. Studies rarely paid attention to the contents and specific characteristics of short video displays and their impacts on consumers' purchase intentions and product evaluations. In fact, the short video displays of products that online retailers offer can be mismatched with consumers' expectations, which may have an influence on consumers' perceptions of the quality of products and purchase intentions. This paper studies the information in short video displays from this perspective. It has practical value for online retailers and consumers. Table 1 provides a literature review.

China's online retail sales reached 9000 trillion yuan in 2018, an increase of 23.9% over the previous year. Clothes occupied 70% of online retail sales, ranking first. As a kind of product that consumers experience, clothes shared in the benefits of online shopping early on. A dynamic display makes consumers provide better product reviews [13]. Thus, this study was designed to explore the short video displays on the Taobao platform. The short video displays' impacts on consumers' perceptions of quality were explored via questionnaires and eight groups of scenario experiments. The study's contributions to and innovations for E-business product displays are as follows: (1) an investigation of a number of online retailers on the Taobao platform, and a summary of the main features of short video product displays on this dominant online retail platform in China; (2) a discussion of the short video displays' impacts on consumers' perceptions of quality in terms of different characteristics, including length, display angle, and usage scenarios; (3) suggestions and proposed solutions for online clothes retailers based on asymmetries between the characteristics of short video product displays that online retailers provide and those that



consumers expect. Our results may help to create a better environment for online clothes shopping and improve the efficiency and quality of online clothes shopping.

**Table 1.** Literature review.

| | Author | Year | Content |
|---|---|---|---|
| Online product displays | Li et al. [8] | 2002 | Consumers' presence and virtual experiences were explored. It was found that three-dimensional (3D) advertisements can enhance consumers' presence and improve consumers' perceptions and purchase intentions. |
| | Khakimdjanova and Park [9] | 2005 | The authors suggest that a product display should be analyzed in five respects, namely, display methods, display techniques, supplementary displays, display aesthetics, and structures and layouts of displays. |
| | Jiang and Benbasat [10] | 2007 | Based on the interactivity of the internet, businesses can use a variety of forms to display their products. The vividness and interactivity of a product display are the main design features that affect the impact of an online product display. |
| | Kim and Lennon [11] | 2010 | The authors found that image magnification technologies have an impact on consumers' enjoyment, which is positively correlated with the perceived quantity of information. |
| | Jungmin and Minjeong [12] | 2014 | The psychological perceptions that are caused by the elements of a product display, such as pictures and text, are the main sources of information with which consumers make purchase decisions. |
| | Wirth et al. [19] | 2016 | Consumers require different information and experience environments for different products, and retailers should adjust the design of their online product displays accordingly. |
| The impacts of online video product displays on consumers' perceptions | Jiang and Benbasat [20] | 2007 | The authors investigated the picture, video, and virtual experience display methods, and they found that both video displays and virtual product experiences had a greater impact on consumers' perceptions than pictures. |
| | Li and Meshkova [21] | 2013 | Product videos and virtual product experiences increased the amount of information consumers received about the tested products and their excitement about the shopping experience. |
| | Roggeveeen et al. [22] | 2015 | Compared with a static display, videos can increase the impact of displayed content and effectively enhance consumers' perceptions of products' value. |
| | Oru et al. [23] | 2016 | The authors discussed the impacts of online product displays' presentation and characteristics on information processing, consumers' attitudes toward products, and purchase intentions. |
| | Flavián et al. [24] | 2017 | Video displays can influence consumers' perceptions and purchase intentions in two respects: ease of imagining products and satisfying demand for tactile sensations. |
| | Guo et al. [25] | 2019 | The usefulness and comprehensiveness of short videos were found to have positive effects on consumers' virtual sense of touch, sense of pleasure, and sense of trust. |

## 2. Pre-Investigation

### 2.1. The Characteristics and Current Status of Short Video Clothes Displays

We found some characteristics of short video displays by sorting and summarizing 100 clothes retailers' short videos as shown in Figure 1. The shortest video was 10 s in length, and the longest video was 60 s in length. Furthermore, 12 retailers did not adopt living models; instead, products were displayed in flat and T-stage ways. Ninety-eight retailers' videos contained background music; however, some retailers used the same music for multiple products, and mismatches between the style of music and the clothing style were found to exist. Fifty-four retailers highlighted their brand in the video. Seventy-seven retailers displayed overall product information. Generally, videos that did not contain a model covered more of the details of products. Analyzed in terms of video length, 30 s was found to be the threshold value; videos longer than this were considered to be long videos, and videos shorter than this were considered to be short videos. Eighty-eight percent of shops displayed short videos. The majority of shops used living models and fashionable background music, and they highlighted the brand at the start or the end of a video. The majority of shops displayed overall product information, then covered the product both overall and in detail.

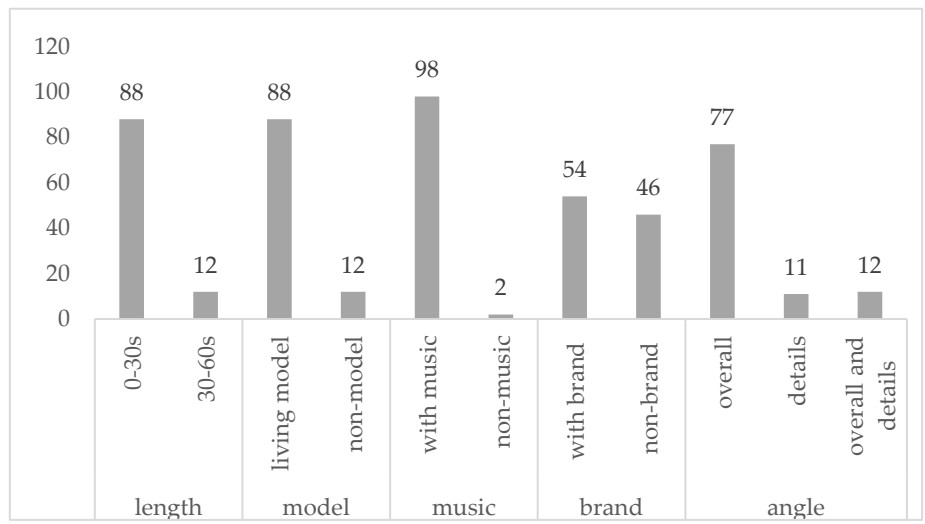

**Figure 1.** The main characteristics of the short video displays.

### 2.2. Consumers' Preferences for the Characteristics of Short Video Clothes Displays

Based on our summary of the characteristics of short video displays, questionnaires were used to pre-investigate consumers' attitudes and preferences for short video clothes displays, so as to provide a basis for scenario experiments. An online questionnaire system was used to distribute and collect questionnaires. A total of 266 questionnaires were collected, of which 19 were invalid. Males accounted for 45.4% of the 207 valid questionnaires, and females accounted for 54.6% of the 207 valid questionnaires. The total average age was 23.4. A total of 73.9% of the respondents had more than three years of online shopping experience.

The statistical results of the questionnaires reveal that, compared with displays without a model, 70.0% of respondents were more likely to watch a video display with a living model, 67.1% of respondents preferred short video displays, 66.7% of respondents were more likely to watch videos with background music, 64.3% of respondents preferred overall product displays shot outdoors, and 51.2% of respondents prefer brand displays. Details are shown in Figure 2.

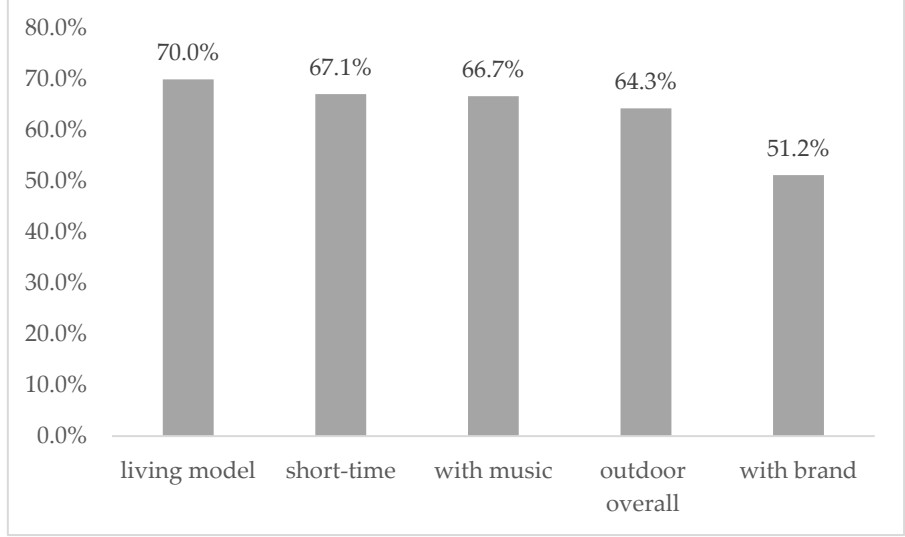

**Figure 2.** The ratios of preferences for the characteristics of short video clothes displays.

From the results of the questionnaires, we can see that the suggestions of respondents focused on the duration and angles of short video displays. Regarding duration, some of the consumers believed that video displays should not only be short in duration, but should also efficiently display information; however, some consumers believed that duration is irrelevant to video displays, which should have an abundance of content and be creative. Regarding angles, some consumers stated that they needed more details and a more comprehensive display.

Our pre-investigation only obtained consumers' preferences for the characteristics of short video displays. We could not use it to determine what kinds of short videos could improve consumers' perceptions of quality. Thus, the result represents only an analysis of consumers' preferences. Further experiments are required to identify the impacts of different video display characteristics on consumers' perceptions of quality.

## 3. Hypotheses

In e-commerce, the vividness of information about a clothing product is enhanced through online short video clothing displays. A short video display is more vivid and emotionally attractive than traditional product display forms [26], such as text descriptions and static picture displays, because a short video display of a product describes the product more specifically and more realistically. The elements of a short video display of a product can be divided into technology, content, aesthetics, and structural layout [27]. The vividness of the information about a product in a short video display is mainly determined by technology and content. From a technical perspective, the vividness of information is determined by the number of senses that are stimulated, the quality of the information, and resolution of the video display [17,28,29]. Short video presentations can stimulate more senses (e.g., visual and auditory rather than visual only). From a content perspective, rich and dynamic images are more likely to attract a consumer's attention and stimulate a consumer's imagination. Many scholars regard video displays as a highly vivid medium that can affect consumers' perceptions of products [7,10,30,31]. At present, there is little research on the question of how the elements of short video displays are arranged and how different characteristics of short video displays affect consumers' perceptions of products' quality.

The results of the pre-investigation show that consumers' suggestions focused on duration and angle of presentation. Furthermore, consumers mentioned the "indoor" and "outdoor" display forms in the pre-investigation. On this basis, we propose hypotheses in three respects, namely, duration, angle of presentation, and usage scenario.

### 3.1. The Length of Videos and Perceptions of Quality

In the internet age, catching consumers' attention under an information overload is difficult. The short video is a method for disseminating content on the internet. With the popularization of mobile terminals and acceleration of the development of networks, short-form and direct content that is provided quickly is in vogue. The short video, as a carrier of fragmented information, not only provides consumers with vivid information, but can also save consumers time when obtaining information. Erik indicates that 20% of users leave within ~10 s of starting to watch a video, 33% of users leave within ~30 s, and 45% leave within in ~1 m [32]. Compared to lengthy videos, shorter videos are more noticeable to viewers [33].

However, the short video is a medium that carries a large amount of data and often needs to be loaded. Long videos require consumers to wait for more time due to buffering, loading, and browsing. Some delay and even freezing may happen during this process. Waiting will make consumers feel anxious and, as the waiting time increases, their anxiety will increase. Negative emotions can lead to negative consequences, such as increased stress, decreased efficiency, and decreased trust [34]. Delays and freezing may directly influence consumers' short-term attention and perceptions [35,36]. Thus, we propose the following hypothesis:

**H1.** *Compared with long videos, consumers have better perceptions of the quality of products in short videos.*

### 3.2. Angle of Presentation and Perceptions of Quality

The display angle of clothing in a short video can be divided into overall display and detailed feature display [37]. Overall display refers to the overall shape of the clothing, and it is intended to give consumers an overall impression of the clothing. Detailed feature display refers to the display of the details of clothing, including fabrics, patterns, and features. Based on overall priority theory [38,39], consumers' perception of products is processed from the overall level to the level of details. When browsing products online, consumers first require information about a product overall, and their perception of the product overall influences their perception of its details [40].

Consumers do not desire product displays that only present details. This study focuses on the impacts of different angles of product displays that contain both overall information and details of products. The display order was overall information followed by details. Video displays that cover both overall information and details better match with the sequence of consumers' perceptions, and they also better satisfy consumers' demands regarding perceptions of online products [11]. Through eye-tracking experiments, Yunyi et al. found that, with respect to the main forms of clothing display, consumers prefer clothing displays with different angles [41]. Thus, we propose the following hypothesis:

**H2.** *Compared with video displays that only contain overall information, consumers have a better perception of the quality of products in video displays that cover both overall information and details.*

### 3.3. Display Scenarios of Short Videos and Perceptions of Quality

In short videos of clothing, different display scenarios will provide consumers with different perceptions. Outdoor scenes will make consumers feel vibrant, free, and dynamic, studio scenes will make people feel professional, rigorous, and monotonous, and indoor scenes will make people feel warm, relaxed, and comfortable [42]. Yunyi et al. found that an "outdoor background" can provide consumers with more aesthetic visual effects than a "studio background" [41]. Zhao points out that consumers have a close relationship with displays of indoor living environments, which produce a strong lived-in atmosphere, and which provide consumers with a lived-in experience and feeling. Compared with indoor scenarios, buildings and street views not only provide consumers with the same lived-in feel as indoor scenarios, but also a more dynamic feel, and the characters are presented more vividly. Thus, street view scenarios are more attractive to consumers [43]. Studies show that vivid information has a positive influence on consumers' attitude and purchase intentions. Because consumers' product reviews are partially determined by the imagined

use of products in different scenarios, a dynamic representation emphasizes the relationship between products and the environment and interactions and links among consumers [13]. Thus, we propose the following hypothesis:

**H3.** *Compared with indoor scenario displays, consumers have a better perception of the quality of products in outdoor scenario displays.*

## 4. Experiment Design

### 4.1. Objectives of and Products Used in the Experiments

The 43rd Chinese Statistical Report on Internet Development shows that, as of December 2018, most (67.8% of all) internet users were aged from 10 to 39 years, of which users aged from 20 to 29 years accounted for 26.8%. The experiments were designed to mainly include female students in university but also some females born after 1980.

Females born after 1980 and 1990, and males born after 1990 are the dominant consumer groups in the online clothes market, which is the biggest online market. Taking the match between online shopper group and product catalog in the online market into consideration, male consumers focus on digital products, and females form the dominant group for clothes. Thus, a one-piece summer dress for females was selected as an experimental product with consideration of the video shot effect.

### 4.2. Experimental Groups and Experimental Materials

The videos that were used in the experiments were grouped into short videos (14 s) and long videos (44 s). Angles of presentation were classified as overall displays and overall and details displays. Display scenarios were grouped into indoor scenarios and outdoor scenarios. As shown in Table 2, there were eight (2 × 2 × 2) groups of experiments in total.

**Table 2.** Experimental design and groups.

| Operating Variables | | Length of Short Videos | |
|---|---|---|---|
| | | **Short** | **Long** |
| Indoor scenarios | Overall display | Experiment group1 | Experiment group 2 |
| | Overall and details displays | Experiment group 3 | Experiment group 4 |
| Outdoor scenarios | Overall display | Experiment group 5 | Experiment group 6 |
| | Overall and details displays | Experiment group 7 | Experiment group 8 |

As shown in Figures 3 and 4, the experimental materials comprised online images, short videos, and information about the selected one-piece summer dress. We sent eight experimental questionnaire links designed on the questionnaire platform to different participants. The participants were informed that they were going to purchase a dress, so they would go to a domestic shopping platform for selection. Then, the picture information of the dress was displayed, as shown in Figure 3. After that, the participants were told that, in addition to the pictures, there would be a short video display about this dress; some screenshots of the video are shown in Figure 4. Participants watched the short video and completed a questionnaire based on their true feelings. To ensure that other variables remained consistent during the experiment, the content of the eight questionnaires was the same except for the short videos.

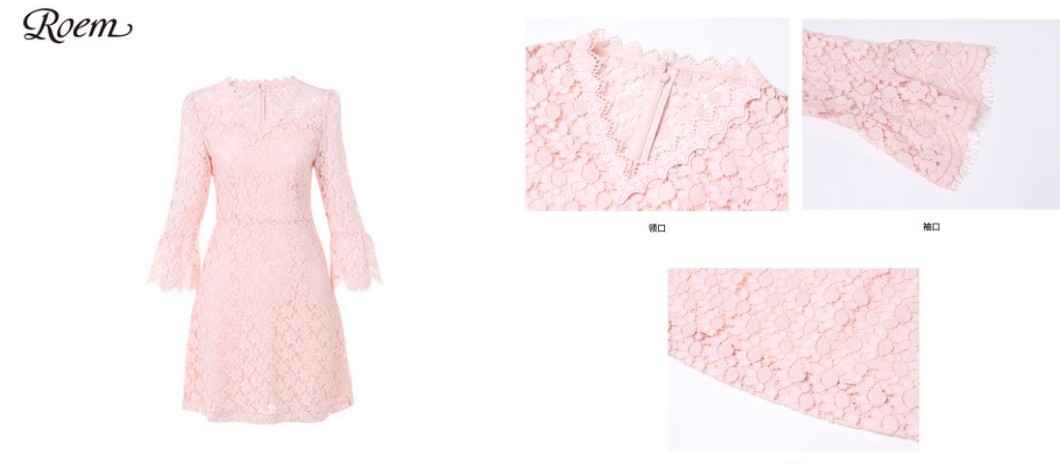

**Figure 3.** Some of the information about the product from the Taobao platform.

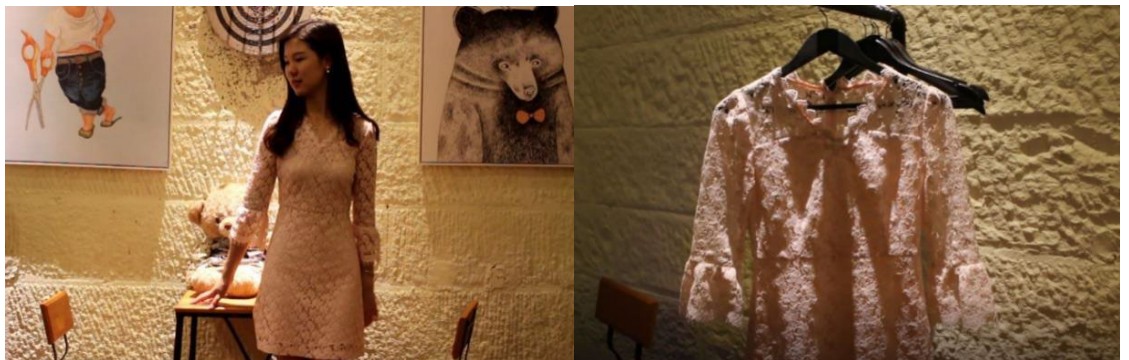

**Figure 4.** Images from the short video display of the product.

### 4.3. Questionnaire Design and Pre-Experiments

Questionnaires were designed to include three parts. The first part collected general information about respondents and their online clothes shopping experiences. The second part contained experimental scenarios, including basic information about the clothes and short videos. The third part contained a measurement table for related variables. The specifications of the questionnaires are shown in Table 3. The existing literature contains little empirical research on the content and characteristics of short videos. Therefore, we referred to a large number of studies on the display of information online, and combined the framework and purpose of this study to design questions through which the respondents' judgements on the characteristics of the short videos could be tested. Then, as shown in Table 3, a measurement scale was designed by reference to studies on perceptions of quality.

A pre-survey was carried out to verify the validity of the questionnaires. Questionnaires for each of the eight group experiments were sent to respondents with a lot of online shopping experience and who had some knowledge about the display of information about clothes online. In total, 48 questionnaires were collected, with six valid questionnaires for each of the experimental groups. The questionnaires were then revised based on the results of the pre-survey.

**Table 3.** The measurement scale.

| Factors | Questions | References |
|---|---|---|
| Judgements on the characteristics of the short videos | The short video is shorter in duration compared to a general short video of clothes.<br>The short video display's angle of presentation is more comprehensive compared with a general short video display of clothes.<br>This short video was shot at an appropriate scene. | Fuxin et al. (2012) [44], Hongxia et al. (2014) [4] |
| Perception of quality | This one-piece summer dress is reliable.<br>This one-piece summer dress is excellently tailored.<br>This one-piece summer dress is of good quality.<br>This one-piece summer dress is durable. | Jarvenpaa and Todd (1997) [45], Baker (1994) [46], Dodds and Grewal (1991) [47] |

## 5. Results

### 5.1. Descriptive Statistics

A total of 335 females were involved in the experiments. The question of "Did you complete this questionnaire seriously?" was included in the questionnaire to help validate the collected questionnaires. A total 249 questionnaires were found to be valid, for a validation rate of 74.33%. SPSS 22.0 was used to process the data. Table 4 shows the initial descriptive statistics for the eight short video groups.

**Table 4.** Descriptive statistics for the eight short video groups.

| Grouping | N | Mean Value | Standard Deviation | Standard Error | Max | Min |
|---|---|---|---|---|---|---|
| Experiment group 1 (indoors, overall, short) | 30 | 3.98 | 1.04 | 0.19 | 6.00 | 2.00 |
| Experiment group 2 (indoors, overall, long) | 30 | 3.43 | 1.25 | 0.23 | 5.00 | 1.00 |
| Experiment group 3 (indoors, overall and details, short) | 33 | 4.45 | 1.06 | 0.18 | 7.00 | 2.25 |
| Experiment group 4 (indoors, overall and details, long) | 31 | 3.99 | 0.86 | 0.15 | 5.75 | 2.50 |
| Experiment group 5 (outdoors, overall, short) | 31 | 4.43 | 1.03 | 0.18 | 6.50 | 3.00 |
| Experiment group 6 (outdoors, overall, long) | 32 | 3.99 | 0.89 | 0.16 | 5.25 | 1.75 |
| Experiment group 7 (outdoors, overall and details, short) | 30 | 4.88 | 0.94 | 0.17 | 6.50 | 3.25 |
| Experiment group 8 (outdoors, overall and details, long) | 32 | 4.45 | 0.79 | 0.14 | 6.00 | 3.00 |

### 5.2. Analysis of Reliability and Validity

The Cronbach's $\alpha$ value was used to measure the reliability of the questionnaires. Table 5 shows the SPSS-determined reliability of the perceived-quality-related questions. The Cronbach's $\alpha$ of perceived quality was 0.924, which is greater than 0.9, indicating that the questionnaires had good reliability. The results of the validity analyses are shown in Tables 6 and 7. The KMO value of perceived quality was 0.843, which lies between 0.8 and 0.9. The factor loading of the four questions was greater than 0.8. This explains 81.25% of the population variance, which indicates that the experimental data have considerable validity.

**Table 5.** Analysis of the reliability of perceived quality.

| Variables | Questions | Deleted $\alpha$ Value | Cronbach's $\alpha$ |
|---|---|---|---|
| Perceived quality | Q1: This one-piece summer dress is reliable. | 0.919 | 0.924 |
| | Q2: This one-piece summer dress is excellently tailored. | 0.892 | |
| | Q3: This one-piece summer dress is of good quantity. | 0.878 | |
| | Q4: This one-piece summer dress is durable. | 0.913 | |

**Table 6.** KMO and Bartlett's sphericity test.

| Variables | KMO Measurement | Bartlett's Sphericity Test | |
|---|---|---|---|
| Perceived quality | 0.843 | Approximate chi-square value | 791.85 |
| | | Df | 6 |
| | | Sig. | 0 |

**Table 7.** The factor loading and the interpretation of total variance.

| | Questions | Factor Loading | The Interpretation of Total Variance |
|---|---|---|---|
| Perceived quality | Q1 | 0.943 | |
| | Q2 | 0.92 | 81.51% |
| | Q3 | 0.88 | |
| | Q4 | 0.866 | |

*5.3. Verification of Hypotheses*

5.3.1. Verification of the Hypothesis on Impacts of the Length of Short Video Displays on Perceptions of Quality

According to the statistical test of significance, an *F*-test was firstly performed for combinations of different lengths and another two factors to determine whether there were significant differences in variance among groups of videos with different lengths. The results show that the *p*-value of each group was greater than 0.05 at the 95% confidence level; thus, this hypothesis cannot be denied. The variance in the long-duration video group and the short-duration video group can be treated as the same. A double-sample equal variance hypothesis *t*-test was used to further analyze whether there were significant differences in the average value of perceived quality between the short-duration video groups and the long-duration video groups. The results of the double-sample equal variance analysis are shown in Table 8. All single-tailed *p*-values of the four groups under the categories of angle and scenario were less than 0.05, which indicates that, when other conditions were held constant, the duration of short videos has a significant impact on consumers' perceptions of quality. After integrating the *t*-values and comparing the average values of the descriptive statistics, the perceived quality of the short-duration video group was found to be significantly better than that of the long-duration video group. Thus, H1 is valid.

**Table 8.** The *t*-tests for the impacts of video displays with different lengths on consumers' perceptions of quality.

| | Duration | *t*-Value | *p* (T ≤ t) Single-Tailed | *t* Single-Tailed Critical |
|---|---|---|---|---|
| Overall and details, outdoors | Long, 44 s | | | |
| | Short, 14 s | −1.989 | 0.026 | 1.671 |
| Overall, outdoors | Long, 44 s | | | |
| | Short, 14 s | −1.799 | 0.038 | 1.670 |
| Overall and details, indoors | Long, 44 s | | | |
| | Short, 14 s | −1.876 | 0.033 | 1.670 |
| Overall, indoors | Long, 44 s | | | |
| | Short, 14 s | −1.849 | 0.035 | 1.672 |

Note: significance level $\alpha = 0.05$.

5.3.2. Verification of the Hypothesis on Impacts of the Angle of Presentation of Short Video Displays on Perceptions of Quality

We used the same verification method for H2 as we used for H1, i.e., an *F*-test followed by a *t*-test. The results of the *F*-test show that the *p*-values of the three groups were greater than 0.05 (at the 95% confidence level), indicating that the populations represented by the three groups have homoscedasticity. The results of the double-sample equal variance hypothesis *t*-test show that the

*p*-value of one of the groups was 0.0024, i.e., less than 0.05, which indicates that the populations represented by this group have heteroscedasticity. As shown in Table 9, the results of the double-sample different variance hypothesis *t*-test show that *p*-values of the four groups were less than 0.05, indicating that there was a significant difference between the two different display angles. After integrating the average values, we concluded that H2 is valid.

**Table 9.** The *t*-tests for the impact of different display angles on consumers' perceptions of quality.

| | Display Angle | *t*-Value | *p* (*T* ≤ *t*) Single-Tailed | *t* Single-Tailed Critical |
|---|---|---|---|---|
| Long, outdoors | Overall and details | | | |
| | Overall | 2.160 | 0.017 | 1.670 |
| Short, outdoors | Overall and details | | | |
| | Overall | 1.801 | 0.038 | 1.671 |
| Long, indoors | Overall and details | | | |
| | Overall | 2.024 | 0.024 | 1.675 |
| Short, indoors | Overall and details | | | |
| | Overall | 1.746 | 0.043 | 1.670 |

Note: significance level $\alpha = 0.05$.

### 5.3.3. Verification of the Hypothesis on Impacts of Short Video Display scenarios on Perceptions of Quality

We used the same verification method for H3 as we used for H1 and H2. The results of the *F*-test show that the *p*-values of the three groups were greater than 0.05 (at the 95% confidence level), indicating that the populations represented by these three groups have homoscedasticity. The results of the double-sample equal variance hypothesis *t*-test show that the *p*-value of one of the groups was 0.0024, i.e., less than 0.05, which indicates that the populations represented by this group have heteroscedasticity. Table 10 shows the results of the double-sample different variance hypothesis *t*-test. The *p*-values of the three groups were smaller than 0.05, and the *p*-value of one of the groups was greater than 0.05. These results indicate that there were significant differences between the indoor scenario and the outdoor scenario. After integrating the average values, we concluded that H3 is valid.

After data processing and analysis, all proposed hypotheses were found to be valid.

**Table 10.** The *t*-tests for different scenarios' impacts on consumers' perceptions of quality.

| | Display Angle | *t*-Value | *p* (*T* ≤ *t*) Single-Tailed | *t* Single-Tailed Critical |
|---|---|---|---|---|
| Long, overall and details | Outdoors | | | |
| | Indoors | 2.180 | 0.017 | 1.670 |
| Short, overall and details | Outdoors | | | |
| | Indoors | 1.718 | 0.045 | 1.670 |
| Long, overall | Outdoors | | | |
| | Indoors | 2.017 | 0.024 | 1.675 |
| Short, overall | Outdoors | | | |
| | Indoors | 1.673 | 0.050 | 1.671 |

Note: significance level $\alpha = 0.05$.

## 6. Conclusions

### 6.1. Research Conclusions

This study explored the impacts of short video displays on consumers' perceptions of quality in an online shopping context. Consumers' preferences were obtained via a survey based on a summary of the characteristics of short video clothes displays. Eight groups of experiments were designed according to video length, angle of presentation, and usage scenario. The impacts of

different video displays on consumers' perceptions of quality were discussed. Our results show that, compared with long-duration video displays, short-duration video displays can make consumers have better perceptions of quality; furthermore, compared with videos that only contain overall information about a product, video displays that contain both overall information and details of a product can make consumers have better perceptions of quality.

*6.2. Theoretical Contributions*

The theoretical contributions of this research are threefold. Firstly, in terms of modern marketing theory, with the development of marketing theory and marketing practice, an interaction-oriented theory centered on customer conceptions was gradually introduced [48,49]. The rapid development of information technology increased the number of opportunities that businesses have to interact with consumers [50], as well as increased consumer demand for product information. Based on modern marketing theories, we explored consumers' subjective perceptions of clothing in short video displays with different characteristics. Previous studies discussed the impact of short video displays on consumers' evaluations in terms of the vividness of the presentation, the ease of imagining products, etc. However, in this study, we specifically explored the impact of different characteristics and attributes of short video displays on consumers' perceptions and evaluations by selecting characteristics of short videos and designing short videos to optimize the marketing strategy for and arrangement of content in short clothing videos based on consumers' evaluations and feedback. Secondly, in terms of cognitive psychology, we used and developed research on the theory of overall priority [38,39]. Research on the overall priority of perceptions is a hot topic in the field of cognitive psychology. Most scholars discuss the effect of overall priority through static pictures [51–55]. This study applied overall priority theory to dynamic images in short videos. Through experiments, it was found that consumers prefer short videos with an overall view that also display details, and there remains an overall priority effect.

Finally, in terms of research on the classification of characteristics of online clothing displays, the five-dimensional classification standard proposed by Khakimdjanova and Papk comprehensively summarizes the characteristics of online clothing displays [9]. Based on the collation and investigation of the content of short video displays from 100 clothing retailers on the Taobao platform, we applied the five-dimensional classification standard to the characteristics of short video displays of clothing, expanding the scope of application of the five-dimensional classification standard.

*6.3. Management Implications*

Currently, more and more online clothes retailers are adopting short videos to display information about their products. The short video is an efficient scenario expression tool. If it is used as a carrier of high-quality content, it can not only provide better pre-purchase experiences [6,22,23], but also increase the conversion rate [25]. However, if the content and structure of short videos are not properly designed, and if they do not satisfy consumers' demands, the value of short video displays will not be realized. For network marketing managers (or e-commerce operators), the question of how to better arrange and design the characteristics of short videos so as to make full use of the ability of short videos to vividly display a product is particularly important [10,31]. According to the results of this research, videos that cover both overall information and details of products, that are short in duration, and that are based on outdoor scenarios can make consumers have a better perception of products' quality. Therefore, three management implications can be drawn. Firstly, businesses should try to reduce the length of videos in order to meet the demand for provision of useful product information in a short time. Secondly, businesses should follow the rule of displaying overall information first, then presenting the details of products, as moving from the overall perspective to the perspective of details can make consumers have a better understanding of products. Thirdly, outdoor scenario displays are closer to real life, and this form of presentation is closer to real product usage scenarios. Hence, businesses should try to use outdoor scenario displays. However, in practice, a decision on whether to use an outdoor scenario display or an

indoor scenario display should take into consideration weather, cost, etc. Therefore, businesses that cannot use outdoor scenario displays should try to utilize props and filming tricks to help make the usage scenario closer to real life and produce an "immersive" atmosphere for consumers [18].

*6.4. Limitations and Prospects*

This study had some limitations. Firstly, we used a limited number of experimental products and samples. As we selected a one-piece summer dress as the experimental product, all respondents were correspondingly female, and the sample size was small. The product type catalogs and applicable groups did not have universality; thus, the scope of applicability of our study's results was limited. Secondly, some unreasonable factors may have been included as characteristics of short videos as summarized by the authors. Thirdly, there were some control variables in and operating limitations to our experiments. Duration, angle of presentation, and usage scenarios of videos were the control variables. However, the design and filming of the short videos were not done professionally. Thus, we were unable to ensure that consumers' perceptions of quality would not be affected by other factors, such as the clothing style and video background preferences of consumers.

Future research could design short videos and perform experiments with other contents and elements of short videos. In this study, three characteristics of short videos were selected to design the short video groups. However, short videos contain more than three elements. The characteristic elements of short videos include technology, content, aesthetics, structure, and layout [27]. Future studies could discuss the characteristics and elements of different short videos based on the design ideas and experimental methods of this study.

Short videos can affect consumers' perceptions through characteristics of the information they present. For example, the comprehensiveness, usefulness, and ease of use of information in short videos will affect consumers' virtual sense of touch, pleasure, trust, and willingness to buy [25]. Therefore, the design of short videos could be based on aspects and factors that do not have an impact on perceived quality, such as the impact of a short video's design on consumers' intention to purchase.

Short videos are time-consuming to load and browse [32]. Although they may provide consumers with more specific information, consumers may also stop waiting to learn about a product by browsing through pictures and reviews. Therefore, one future research direction is to start with the negative impacts of short videos, in order to provide businesses with more ideas to help them design better short videos.

In addition, we selected a woman's summer dress in the clothing category as the experimental object. Future research could strengthen our scientific and general manipulation of the selection of experimental objects, as well as apply this research to other fields, such as food, cosmetics, and other products that consumers experience.

**Author Contributions:** Conceptualization, B.S. and D.D.; methodology, R.M.; software, J.C.; validation, R.M., D.D., and B.S.; formal analysis, D.D.; investigation, J.C.; data curation, D.D.; writing—original draft preparation, D.D. and B.S.; writing—review and editing, B.S. and D.D.; funding acquisition, R.M. and B.S. All authors have read and agreed to the published version of the manuscript.

**Funding:** This research was funded by the Chinese National Funding of Social Sciences (No. 14AGL023) and the Fundamental Research Funds for the Central Universities (No. 2019 CDJSK 02 PT 19).

**Acknowledgments:** We thank all commenters for their valuable and constructive comments.

**Conflicts of Interest:** The authors declare no conflicts of interest.

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
