# Peer review of "The Impacts of Online Clothes Short Video Display on Consumers’ Perceived Quality"

_information, doi:10.3390/info11020087_

Round 1

Reviewer 1 Report

Dear author(s),

Thank you very much for the opportunity to review your manuscript. The paper has some contributions in application area and online clothes could benefit from it. However, the paper is not up to the desired standards and you should revise it considering the below comments:

How to design the questionnaire or survey in this study? Every questionnaire design should start with a review of the existing literature or surveys with similar research topics. In my opinion, four questions cannot reflect on the practical acts of experimental subjects unless the researcher provides a reasonable basis. What is “significant analysis method”? Please explain. Please explain better your research contributions with previous studies in Conclusions section. Improve English and grammar.

Sincerely.

Reviewer 2 Report

Videos play an increasing role in online marketing, so more research about videos is highly relevant.

The current study has the potential to contribute to this discussion.

Here are my concerns:

The theory and hypotheses section are quite short. Overall, the paper is really short, so there is enough "room" for improvements. Another example: the theoretical implication section is just few sentences long. If there is not more to say about the contribution, the paper is not publishable. so please rethink how your paper adds to the literature.

In the discussion section, I would like to see some more "futuristic" outlooks. What comes in the era "after vFor instance, look at Dr Felixs research on AR content how it can inspire consumers (https://www.sciencedirect.com/science/article/pii/S0969698918310257 ) Do you expect your findings to be appplicable or different for these new developments?

The development of the hypotheses is weak. For instance, you build on overall priority theory in H2. This theory deserves much more attention in your paper.

249 respondents for 8 groups seems a bit too small.

I would revise the methods section. It reads more like a student paper than like an academic paper.

I also recommend the authors to use a professional copy editor.

Round 2

Reviewer 1 Report

The author(s) has satisfactorily responded to all my questions and made the necessary changes to the manuscript. Note, please check if typing of your paper follows the journal format.

Regards,